# Dairying enabled Early Bronze Age Yamnaya steppe expansions

Shevan Wilkin[1,2 ✉], Alicia Ventresca Miller[1,3], Ricardo Fernandes[1,4,5], Robert Spengler[1], William T.-T. Taylor[1,6], Dorcas R. Brown[7], David Reich[8,9,10,11], Douglas J. Kennett[12], Brendan J. Culleton[13], Laura Kunz[14], Claudia Fortes[14], Aleksandra Kitova[15], Pavel Kuznetsov[16], Andrey Epimakhov[17,18], Victor F. Zaibert[19], Alan K. Outram[20], Egor Kitov[21,22], Aleksandr Khokhlov[16], David Anthony[7,11] & Nicole Boivin[1,23,24,25 ✉]

During the Early Bronze Age, populations of the western Eurasian steppe expanded across an immense area of northern Eurasia. Combined archaeological and genetic evidence supports widespread Early Bronze Age population movements out of the Pontic–Caspian steppe that resulted in gene flow across vast distances, linking populations of Yamnaya pastoralists in Scandinavia with pastoral populations (known as the Afanasievo) far to the east in the Altai Mountains[1,2] and Mongolia[3]. Although some models hold that this expansion was the outcome of a newly mobile pastoral economy characterized by horse traction, bulk wagon transport[4–6] and regular dietary dependence on meat and milk[5], hard evidence for these economic features has not been found. Here we draw on proteomic analysis of dental calculus from individuals from the western Eurasian steppe to demonstrate a major transition in dairying at the start of the Bronze Age. The rapid onset of ubiquitous dairying at a point in time when steppe populations are known to have begun dispersing offers critical insight into a key catalyst of steppe mobility. The identification of horse milk proteins also indicates horse domestication by the Early Bronze Age, which provides support for its role in steppe dispersals. Our results point to a potential epicentre for horse domestication in the Pontic–Caspian steppe by the third millennium BC, and offer strong support for the notion that the novel exploitation of secondary animal products was a key driver of the expansions of Eurasian steppe pastoralists by the Early Bronze Age.

The pastoralist populations of the Eurasian steppe have long been a source of archaeological and historical fascination. Although the later history of steppe pastoralists—including the rise of the Xiongnu and Mongol empires in the east—is reasonably well-established, the early emergence and expansion of pastoralist groups in the steppe occurred before the historical era and has largely been reconstructed on the basis of archaeological and linguistic data[1,6,7]. More recently, ancient DNA evidence has provided insights into early steppe populations, revealing evidence for a major influx of steppe ancestry into Europe in the Late Neolithic that effectively transformed the European genetic landscape[1,2,8]. Archaeogenetic data also link these same populations (referred to as Yamnaya) with pastoral Afanasievo populations far to the east in the Altai Mountains[1,2] and Mongolia[3]. Combined archaeological and genetic evidence supports widespread population movements in the Early Bronze Age (about 3300 to 2500 BC) from the Pontic–Caspian steppe that resulted in gene flow across vast distances, linking Yamnaya pastoralist populations in Scandinavia with groups that expanded into Siberia[9].

Although the Yamnaya expansions are well-established, the driving forces behind them remain unclear. A widely cited theory holds that the early spread of herders across Eurasia was facilitated by a newly mobile pastoral economy that was made possible by a combination of horse traction and bulk wagon transport[4–6]. Together with regular dietary dependence on meat and milk[5], this opened up the steppe to exploitation and occupation by pastoralist communities. Yet for all its persuasiveness, the model remains inadequately supported by direct

[1]Department of Archaeology, Max Planck Institute for the Science of Human History, Jena, Germany. [2]Institute for Evolutionary Medicine, Faculty of Medicine, University of Zürich, Zürich, Switzerland. [3]Department of Anthropology, University of Michigan, Ann Arbor, MI, USA. [4]School of Archaeology, University of Oxford, Oxford, UK. [5]Faculty of Arts, Masaryk University, Brno-střed, Czech Republic. [6]Department of Anthropology, University of Colorado, Museum of Natural History, Boulder, CO, USA. [7]Department of Anthropology, Hartwick College, Oneonta, NY, USA. [8]Department of Genetics, Harvard Medical School, Boston, MA, USA. [9]Broad Institute of Harvard and MIT, Cambridge, MA, USA. [10]Howard Hughes Medical Institute, Harvard Medical School, Boston, MA, USA. [11]Department of Human Evolutionary Biology, Harvard University, Cambridge, MA, USA. [12]Department of Anthropology, University of California, Santa Barbara, CA, USA. [13]Institutes of Energy and the Environment, The Pennsylvania State University, University Park, PA, USA. [14]Functional Genomics Centre Zürich, University of Zürich/ETH, Zürich, Switzerland. [15]Center for Egyptological Studies, Russian Academy of Sciences, Moscow, Russian Federation. [16]Samara State University of Social Sciences and Education, Samara, Russian Federation. [17]South Ural State University, Chelyabinsk, Russian Federation. [18]Institute of History and Archaeology, Ural Branch of the Russian Academy of Sciences, Yekaterinburg, Russian Federation. [19]Institute of Archaeology and Steppe Civilizations, Al-Farabi Kazakh National University, Almaty, Kazakhstan. [20]Department of Archaeology, University of Exeter, Exeter, UK. [21]Center of Human Ecology, Institute of Ethnology and Anthropology, Russian Academy of Sciences, Moscow, Russian Federation. [22]Faculty of History, Archaeology, and Ethnology, Al-Farabi Kazakh National University, Almaty, Kazakhstan. [23]School of Social Science, The University of Queensland, Brisbane, Queensland, Australia. [24]Department of Anthropology and Archaeology, University of Calgary, Calgary, Alberta, Canada. [25]Department of Anthropology, National Museum of Natural History, Smithsonian Institution, Washington, DC, USA. ✉e-mail: wilkin@shh.mpg.de; boivin@shh.mpg.de

# Article

archaeological or biomolecular data. Archaeological evidence for the use of bulk wagon transport by the Eneolithic Maikop and Early Bronze Age Yamnaya groups exists in the form of carts and bridling materials[10], but two other critical components of the model—a reliance on domesticated horses and ruminant dairying—remain archaeologically unproven.

The domestication status of Eurasian horses has long been debated[5,6,11–14], and recent archaeogenetic findings[15] have shifted our understanding of early horses at the Eneolithic site of Botai in northern Kazakhstan by identifying them as *Equus przewalskii* rather than the modern-day domestic horse (*Equus caballus*)[5,6]. Although horses do appear in Early Bronze Age assemblages on the steppe, it remains unclear whether they were being ridden[7,11,16,17], or indeed whether they were part of pastoral herds or simply hunted. On the eastern Eurasian steppe, growing evidence suggests that horses were not ridden[11,12,18] or milked[19] before about 1200 BC, and horses may have been uncommon in early pastoralist assemblages[20]. Early ruminant dairying on the western steppe has also been inadequately demonstrated, as human stable isotope data from the region suggests—but cannot confirm—dairy consumption[21,22]. Palaeoproteomics, which is the only method that is able to evince individual dairy consumption (rather than milk production) and provide taxonomic resolution, has so far been minimally applied to steppe populations. Across Yamnaya and Afanasievo populations, dairying evidence is available only for a few individuals from the eastern steppe who have ancestry from western steppe groups; the earliest individual provides only a taxonomically ambiguous ruminant (*Ovis/Bos*) peptide result[19].

To address the heavily debated question of what drove Yamnaya expansions across the steppe[6,23–25], we conducted proteomic analysis of dental calculus sampled from 56 steppe individuals who span the Eneolithic to Late Bronze Age, and who date from between 4600 and 1700 BC. Our samples from the Eneolithic (about 4600 to 3300 BC) are from 19 individuals from 5 sites: Murzikha 2 (6 individuals), Khvalynsk 1 and Khvalynsk 2 (9 individuals), Ekaterinovka Mys (1 individual), Lebyazhinka 5 (1 individual) and Khlopkov Bugor (2 individuals) (Fig. 1, Supplementary Fig. 1a). Ancient DNA results from Khvalynsk and other Eneolithic sites in the Volga and northern Caucasus[2,7,26] support the existence of an Eneolithic population across this region that was genetically similar to the Yamnaya population, but who lacked the additional farmer (Anatolian) ancestry that would arrive later on the steppe[7]. Published stable isotope and archaeological studies applied to Eneolithic populations from the Pontic region point to an economy based on fishing, the gathering of local plants and the keeping of domesticated animals[6,21,27,28]. Given the importance of the horse in reconstructions of early pastoralist expansions, we also examined dental calculus from two individuals from the well-known site of Botai. With faunal assemblages dominated by horse remains[11–13] and early lipid studies of ceramics indicating horse milking at the site by 3500 BC[13], the site is central to discussions of early horse milking and dairying in the Eurasian steppe.

Our Bronze Age samples come from 35 individuals from 20 sites in the Volga–Ural steppes that can be divided into two chronological groups: the Early Bronze Age (about 3300 to 2500 BC) era of Yamnaya-culture mobile pastoralism[29,30]; and the Middle–Late Bronze Age transition (about 2500–1700 BC), when chariots, fortified settlements and new western-derived influence genetic ancestries appeared with the Sintashta culture[31]. The cemetery sites and the number of individuals (in parentheses) from the Early Bronze Age are: Krasikovskyi 1 (2) Krasnokholm 3 (1), Krivyanskyi 9 (2), Kutuluk 1 (2), Leshchevskoe 1 (1), Lopatino 1 (1), Mustayevo 5 (2), Nizhnaya Pavlovka (1), Panitskoe (1), Podlesnoe (1), Pyatiletka (1) and Trudovoy (1); and, from the Middle–Late Bronze Age transition, Bolshekaraganskyi (1), Kalinovsky 1 (2), Kamennyi Ambar 5 (3), Krasikovskyi I (1), Krivyanskiy 9 (3), Lopatino 1 and Lopatino 2 (2), Potapovka 1 (1), Shumayevo 2 (1) and Utevka 6 (5) (Supplementary Fig. 1b, c). Archaeological and stable isotope findings[6,22] indicate that the diet of Early Bronze Age Yamnaya groups was focused on herd animals, specifically cattle, sheep and goat. Horse remains also appear in quantity on a few steppe archaeological sites, but the status of Early Bronze Age horses—whether domesticated or hunted—has remained unclear[32,33]. The Middle–Late Bronze Age transition saw a shift to greater horse exploitation and chariot use, within the context of an ongoing dietary focus on domesticated livestock.

Of the 56 ancient human dental calculus samples we tested, 55 were successfully extracted and produced identifiable protein data. Of these 55, 48 (87%) were determined to have strong signals for preservation through an assessment of proteins commonly found within the oral cavity; detailed information on this assessment is provided in Methods, Supplementary Table 3.

The earliest samples in our study (about 4600 to 4000 BC) are from 5 Eneolithic sites in southwestern Russia located on or close to the Volga River and its tributaries. Of the samples from these 19 individuals, 11 were successfully extracted and well-preserved, and 10 of these did not show any evidence for dairy consumption (Figs. 1a, 2a). The calculus of one individual contained two peptides specific to bovine (*Bos*, *Bubalis* and *Bison*) α-S1-casein, a milk curd protein. However, as the only dietary peptides contained in this sample were specific to casein and evidence for the most commonly recovered dairy protein β-lactoglobulin (BLG) was lacking, dairy consumption in this individual could not be confidently confirmed. In general, casein peptides appear to preserve more poorly than BLG in archaeological calculus, and thus are most often identified together with other dairy protein peptides rather than alone[19,34–37]. Additionally, within the two identified casein peptides, there is only one possible amino acid deamidation site, which renders any estimation of the antiquity of these peptides exceedingly challenging. A previously published paper[38] demonstrates the extreme variability in deamidation of amino acids in milk proteins, which further limits our ability to confirm the authenticity of this dairy finding. The calculus from the two additional Botai individuals demonstrated adequate preservation, but also lacked evidence for dairy consumption.

For the Early Bronze Age individuals (dating to the onset of the Yamnaya cultural horizon), dairy peptides were recovered from 15 of the 16 individual calculus samples we analysed (Fig. 1b, 2b). All 15 individuals with positive dairy results contained multiple peptide spectral matches to ruminant dairy proteins (including BLG), and some individuals also contained α-S1 casein, α-S2-casein or both. Although many of the milk peptides were only specific to higher taxonomic levels (such as Pecora, an infraorder within Artiodactyla (cow, sheep, goat, buffalo, yak, reindeer, deer and antelope)), others enabled more specific taxonomic classifications, including to family, genus or species. We found *Ovis*, *Capra* and *Bos* attributions, and the calculus of many individuals contained dairy peptides from several species. Notably, we identified *Equus* milk peptides from the protein BLGI in 2 of 17 Early Bronze Age individuals, both from the southwestern site of Krivyanskiy 9 (3305 to 2633 calibrated years BC (Supplementary Table 5 provides individual accelerator mass spectrometry dating information)). Although the genus *Equus* includes horse, donkey and kiang, only horse species (*E. caballus, E. przewalskii, Equus hemionus* and *Equus ferus*) are archaeologically attested in the steppe in the Early Bronze Age, supporting the *Equus* identification as horse.

For the Middle–Late Bronze Age transition, calculus samples from 15 of 19 individuals were positive for evidence of ruminant milk consumption (Figs. 1c, 2c). Similar to the Early Bronze Age, we identified BLG, α-S1-casein and α-S2-casein, as well as the whey protein α-lactalbumin. Taxonomic identifications again ranged from the Pecora infraorder to genus-level identifications (including *Ovis* and *Bos*), but without any specific identifications for *Capra* or *Equus*. Supplementary Table 4 provides a full accounting of all identified dairy proteins for each individual.

Overall, our results point to a clear and marked shift in milk consumption patterns between the Eneolithic and Early Bronze Age in the Pontic–Caspian Steppe. The majority of Eneolithic individuals (10 out of 11 (92%)) in our assemblage lack any evidence for milk consumption,

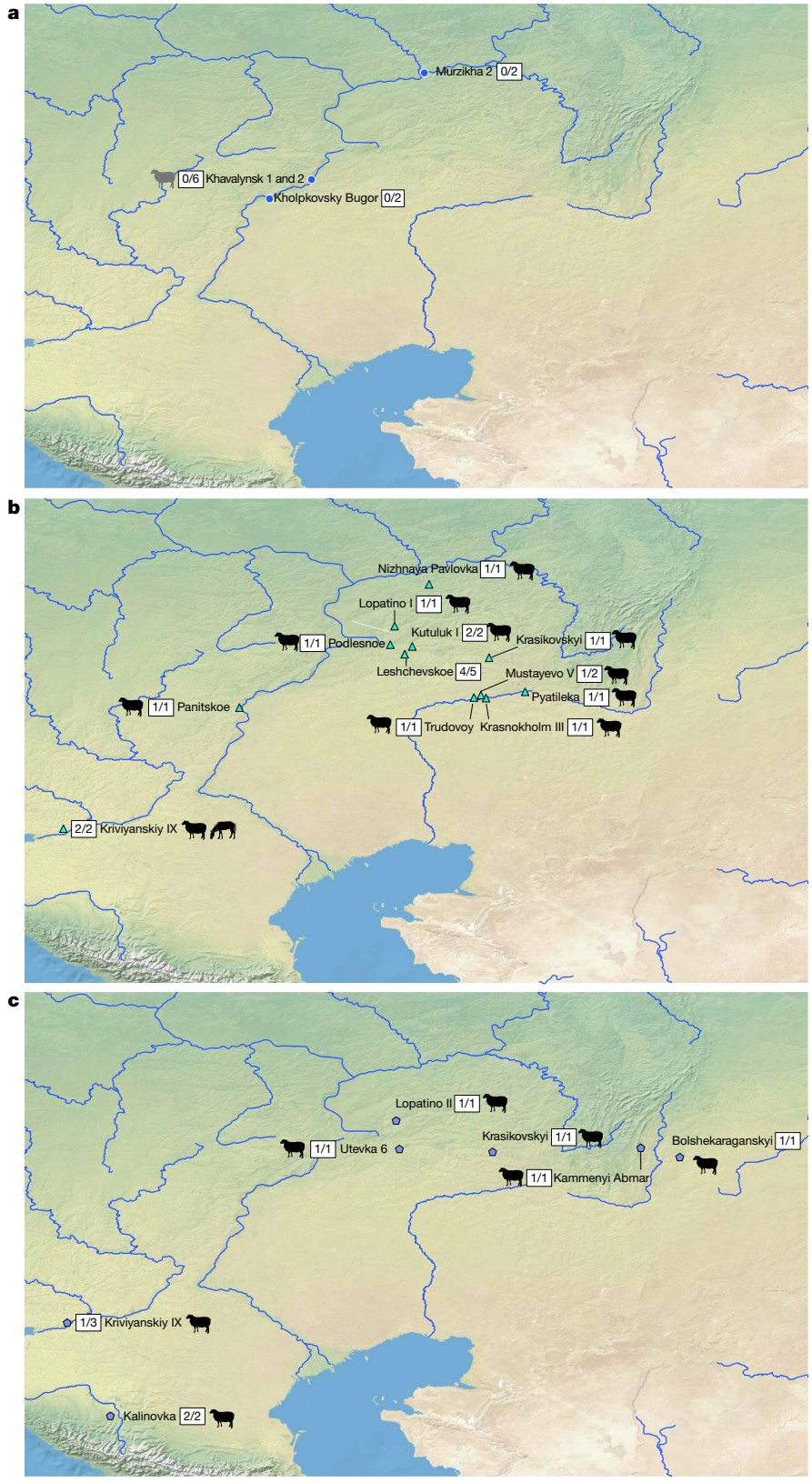

**Fig. 1 | Map showing sites that yielded individuals with preserved ancient proteins. a–c**, Eneolithic (**a**), Early Bronze Age (**b**) and Middle–Late Bronze Age (**c**) sites in the Pontic–Caspian region, showing the number of individuals with a positive dairy identification out of the total number of individuals with preserved ancient proteins for each site. Strong evidence of preservation of equine or ruminant milk protein identifiers are depicted with black animal icons; the single individual with equivocally identified casein peptides is shown with a grey icon. For a map of all sites (including those without preserved proteins), see Supplementary Fig. 1. Base maps were created using QGIS 3.12 (https://qgis.org/en/site/), and use Natural Earth vector map data from https://www.naturalearthdata.com/downloads/. The horse image is reproduced from ref. [33]; sheep silhouette, public domain (https://thenounproject.com/icon/12538/).

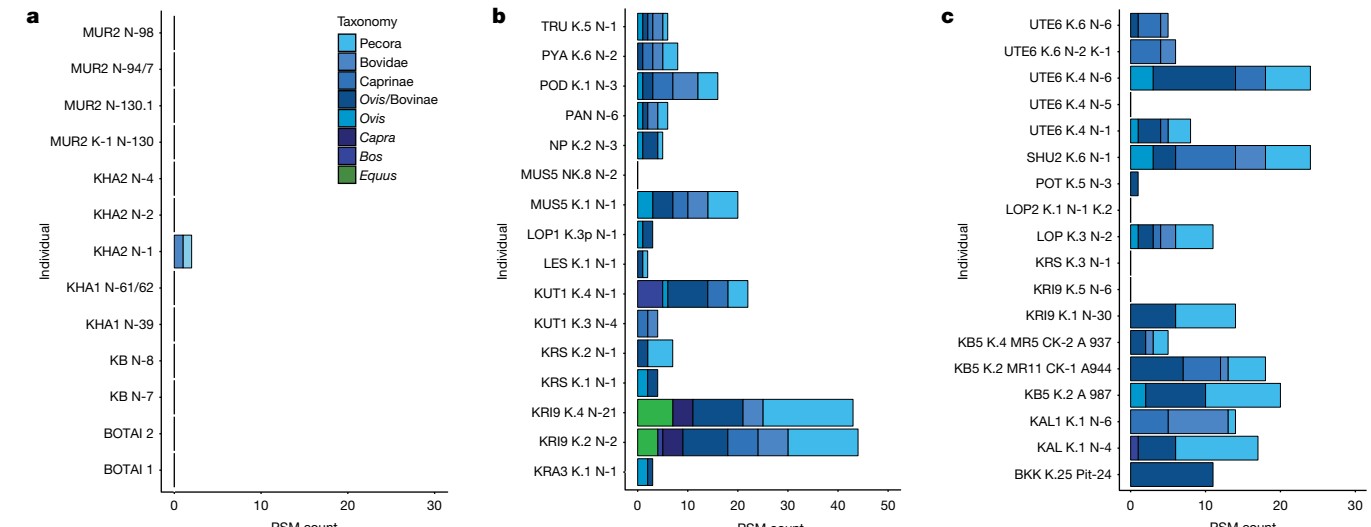

**Fig. 2 | Histogram of taxonomic specificity of dairy peptide spectral matches per individual. a–c**, Histograms for individuals with evidence for consumption of dairy, from the Eneolithic (**a**), Early Bronze Age (**b**) and Middle and Late Bronze Age (**c**). PSM, peptide spectral match.

whereas the overwhelming majority of Early Bronze Age individuals (15 out of 16 (94%)) contain ample proteomic evidence for dairy consumption in their calculus. Although a single individual at Eneolithic Khvalynsk with somewhat equivocal evidence for the consumption of dairy from cattle may indicate small-scale dairy use, the reliability of this single identification is questionable. Our findings suggest that regular dairy consumption in the Pontic–Caspian Steppe began only at the time of the Eneolithic-to-Early Bronze Age transition. Although neighbouring Eneolithic farming populations in Europe appear to have been dairying[39], those living across the steppe frontier did not adopt milking practices, which suggests the presence of a cultural frontier. The proteomic data are in broad agreement with findings from lipid analyses in the Ukraine (Supplementary Information section 2, Supplementary Table 2). They also agree with stable isotope analysis of individuals from Eneolithic-to-Bronze-Age Samara showing a corresponding shift from a heavy reliance on fish, deer and other riverine forest (C3) resources to a greater reliance on terrestrial and grassland (C3 and C4) animal products[22,40].

One important advantage of proteomic data is their ability, in some cases, to provide species-specific protein identifications. Our study offers evidence for the Bronze Age milking of sheep, goat and cattle, which fits with evidence for the herding of these animals. The lush valleys of the Pontic–Caspian Steppe provided ample forage and hydration for mixed herds of arid-adapted sheep and goat, as well as more water-reliant cattle[41,42]. Although a recent study has shown that lactase persistence—which results from the presence of an allele that enables production of lactase into adulthood—was rare in steppe populations of the Early Bronze Age[43], we find that the western steppe community was regularly consuming dairy that could have included fresh milk and/or other processed products with reduced lactose, such as yogurts, cheeses or fermented milk beverages.

Our study of dental calculus from the Eneolithic site of Botai to the east, where early horse milking has been suggested by lipid analysis (albeit equivocally[44]), did not yield milk proteins. Although two samples are insufficient for drawing broad conclusions, this finding does not support widespread milk consumption at the site[13,45,46]. However, two calculus samples from Early Bronze Age individuals of the Pontic–Caspian region do provide evidence for the consumption of horse milk. Combined with archaeogenetic evidence[15] that places the Botai horses on a different evolutionary trajectory than the domesticated DOM2 *E. caballus* lineage, this finding—if backed up by further sampling and analysis—would seem to firmly shift the focus of sustained early horse

domestication on the Eurasian steppe to the Pontic–Caspian region. So far, the oldest horse specimens that carry the DOM2 lineage date to between 2074 to 1625 calibrated years BC, at which time the lineage is archaeologically attested in present-day Russia, Romania and Georgia[15]. Our identification of—to our knowledge—the earliest horse milk proteins yet identified on the steppe or anywhere else reveals the presence of domestic horses in the western steppe by the Early Bronze Age, which suggests that the region (where the first evidence for horse chariots later emerged at about 2000 BC[47]) may have been the initial epicentre for domestication of the DOM2 lineage during the late fourth or third millennium BC.

Overall, our findings offer strong support to the notion of a secondary products revolution[48,49] in the Eurasian steppe by the Early Bronze Age. This change in subsistence economy, indicated by dietary stable isotopes in human bones as well as by proteomics, was accompanied by the widespread abandonment of Eneolithic riverine settlement sites, the appearance of kurgan cemeteries in the previously unexploited arid plateaus between the river valleys, and the inclusion of wheeled vehicles and occasional horse bones in Yamnaya graves. At the same time, the steppe Yamnaya population expanded westward into Europe and eastward to the Altai Mountains (a range of 6,000 km)[1,3,50]. Although we cannot offer direct insight into the question of horse riding or traction on the basis of our data, evidence for milked horses certainly makes horse domestication more likely, and may indicate that horses had a role in the spread of Yamnaya groups. The triad of animal traction, dairying and horse domestication appears to have had an instrumental role in transforming Pontic–Caspian economies and opening up the broader steppe to human habitation by the Early Bronze Age. If some or even all of these elements were present before the Bronze Age, it is only from this latter period that we witness their intensive and sustained exploitation amongst numerous groups. Although other factors will no doubt also have been important, the emergence of more mobile, pastoralist societies adapted to survival on the cold and arid steppe—where horses may have opened up snow-covered pasturage for other animals[18], and milk would have provided a sustained source of protein, nutrients and fluids—was undoubtedly critical to the expansion of Bronze Age pastoralists such the Yamnaya groups.

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

# Article

## Methods

No statistical methods were used to predetermine sample size. The experiments were not randomized, and investigators were not blinded to allocation during experiments and outcome assessment.

### Protein extraction and data analysis methods

**Sample collection.** Dental calculus was collected at the Department of Archaeology at Samara State University and the Museum at the Institute of Plant and Animal Ecology at the Ural Branch of the Russian Academy of Sciences. Calculus was collected in sterile tubes and hand-carried to the Max Planck Institute for the Science of Human History (MPI-SHH) in Jena. Calculus from the Botai site was sampled at the site; it was collected in sterile tubes and shipped to MPI-SHH. Each calculus sample was removed using a clean dental scaler, and implements were cleaned with alcohol swabs between the sampling of different individuals. Contamination from modern human keratin and environmental collagen that may have occurred owing to previous sampling for ancient DNA or stable isotope analysis was reduced through the use of nitrile gloves during collection, and samples were taken directly from the teeth into clean, 2-ml Eppendorf tubes, in which they were stored until protein extraction in the Palaeoproteomics Laboratory at the MPI-SHH.

**Protein extraction.** For samples with a 'Z' designation, proteins were extracted using a modified low-volume filter-aided sample preparation that has previously been described[35]. To decrease contamination, 1 ml of 0.5 M EDTA was then added to each sample tube, samples were rotated for 5 min followed by centrifugation at 20,000 rcf for 10 min to remove the any contamination on the outer layer of calculus; the supernatant was then removed and retained. Then, 1 ml of 0.5 M EDTA was added to each decontaminated sample, and the sample was allowed to decalcify on a rotator for 5–7 days at room temperature until completion. After demineralization, samples were centrifuged at top speed (20,000 rcf) for 5 min. Eight hundred μl of EDTA supernatant was removed and stored for future analysis. Separately, 50 μl of urea solution (8 M) was added to a 30-kDa Millipore Microcon filter unit. Samples were denatured, reduced and alkylated with 30 μl of sodium dodecyl sulfate (SDS)-lysis buffer (4% w/v SDS, 100 mM Tris/HCL pH 8.2 and 0.1 M DTT), incubated at 95 °C for 5 min, and 100 μl of iodoacetamide (IAA) solution (0.5 M IAA and 8 M UA) was added to the filter units and was mixed at 600 rpm for 1 min in the dark. Following incubation, the samples were centrifuged at 14,000g for 10 min. Two hundred μl of UA (8 M) was added to the filter unit, followed by the lysed sample supernatant and units centrifuged for 20 min at 14,000g. UA was used twice to remove the IAA and followed by centrifugation for 12–15 min at top speed. One hundred μl of 0.5 M NaCl was added to each filter unit, spun for 12 min at 14,000g. This step was repeated and the flow-through discarded. The filter units were transferred to new units, and 120 μl of trypsin solution (3 μl of 0.4 μg μl⁻¹ trypsin in 117 μl of 0.05 M triethylammoniumbicarbonate) was added to each unit. Units were thermomixed at 600 rpm for 1 min and then incubated overnight at 37 °C. Following digestion, samples centrifuged for 20 min at 14,000g and acidified with 5% TFA to a pH of <2.

Stage tips (Thermo Scientific StageTips 200 μl C18 tips) were cleaned with 100% methanol, followed by 60% acetonitrile (ACN) solution (60% ACN, 0.1% TFA and 39.9% ddH₂O). Then, each was equilibrated with 2 washes of 150 μl of 3% ACN solution (3% ACN, 0.1% TFA and 96.9% ddH₂O). Samples were then loaded onto the tips and twice washed with 3% ACN and 0.1% TFA and the flowthrough discarded. Peptides were collected in new tubes from the stage tip with 150 μl 60% ACN solution and each was dried in an evaporator and stored at −80 °C until liquid chromatography with tandem mass spectrometry (LC–MS/MS) analysis.

Samples with a 'DA' were extracted using a single pot, solid phase enhanced sample preparation (SP3) modified for archaeological dental calculus samples. One millilitre of 0.5 M EDTA was added to each

sample, and samples were then placed on a rotator for 5 min and centrifuged at top speed (20,000 rcf) for 10 min. The entire supernatant was removed and retained, and an additional 500 μl of 0.5 M EDTA for demineralization was added and samples were placed back onto the rotator for 5–7 days.

Following demineralization, samples were centrifuged at 20,000 rcf for 10 min, and 400 μl of the supernatant was removed and retained. To increase denaturation, reduce and alkylate, 200 μl of 6 M guanidine hydrochloride and 30 μl of 40 mM CAA, 100 mM TCEP were added to the pellet and remaining supernatant and mixed through resuspension. Samples were then placed on a heating block (Cell Media, Thermoshaker Pro) and heated to 99 °C for 10 min. Upon removing samples from heat, 20 μl of a 20 μg μl⁻¹ 50/50 mixture of hydrophilic and hydrophobic SeraMag SpeedBeads was added to each sample, and to increase protein–bead adhesion, 350 μl of 100% ethanol was then added to each tube. Samples were then placed on the ThermoMixer for 5 min at 1,000 rpm at 24 °C. Upon removal from the ThermoMixer, tubes were placed on a magnetic rack, which moved the beads to the wall side of the tube. With the proteins now adhering to the beads, the entire supernatant was removed and retained for possible later analysis. To remove any non-proteinaceous materials, 3 washes of 200 μl 80% ethanol were carried out. Once the beads were thoroughly washed, 100 μl of 100 mM ammonium bicarbonate was added to each tube, as well as 0.2 μg of trypsin. Samples were then placed on the ThermoMixer at 37 °C at 750 rpm. After 10 min, samples were resuspended and left on the ThermoMixer overnight (18 h) for protein digestion. Following digestion, sample tubes were centrifuged at 20,000 rcf for 1 min, and then placed back onto the magnetic rack. The entire supernatant was removed and transferred to a clean tube. Each sample was then acidified with 5% TFA to reduce the pH to <2. Acidified sample tubes were again centrifuged at top speed for five minutes to push any remaining non-proteinaceous materials into a pellet and improve stage tip clean up. Stage tips were prepared with 150 μl MeOH, and centrifuged at 2,000 rcf, followed by 60% ACN, 0.1% TFA and another round of centrifugation. To equilibrate the stage tips, we added 150 μl 3% ACN, 0.1% TFA, followed by another centrifugation step, and these steps were then repeated. Samples were added to each stage tip, and centrifuged for 3 min at 2000 rcf, or until the entire sample had passed the stage tip. This was followed by an additional 2 rinse steps with 3% ACN, 0.1% TFA. Samples were not eluted at the MPI-SHH, but retained on stage tips in the −20°C freezer until shipment to the Functional Genomics Center Zürich at the University of Zürich. A full detailed protocol is available at protocols.io (https://doi.org/10.17504/protocols.io.bfgrjjv6).

**High performance LC–MS/MS analysis.** The samples were sent on stage tips to the Functional Genomics Center. There, the peptides were eluted from the stage tips and dried. After resolubilization in 10 μl of 3% ACN, 0.1% formic acid, the peptide level was normalized using the DeNovix DS-11 Series Spectrophotometer.

**LC–MS/MS analysis.** For samples with a laboratory identifier that starts with Z (Supplementary Table 3), mass spectrometry analysis was performed on a Q Exactive HF mass spectrometer (Thermo Scientific) equipped with a Digital PicoView source (New Objective) and coupled to a M-Class UPLC (Waters). Solvent composition at the two channels was 0.1% formic acid for channel A and 0.1% formic acid, 99.9% ACN for channel B. Column temperature was 50 °C. For each sample, 4 μl of peptides were loaded on a commercial ACQUITY UPLC M-Class Symmetry C18 Trap column (100 Å, 5 μm, 180 μm × 20 mm, Waters) followed by ACQUITY UPLC M-Class HSS T3 column (100 Å, 1.8 μm, 75 μm × 250 mm, Waters). The peptides were eluted at a flow rate of 300 nl min⁻¹ by a gradient from 5 to 40% B in 62 min. Column was cleaned after the run by increasing to 98% B and holding 98% B for 5 min before re-establishing the loading condition. Samples were acquired in a given order.

The mass spectrometer was operated in data-dependent mode, acquiring full-scan mass spectra (350–1,500 $m/z$) at a resolution of 120,000 at 200 $m/z$ after accumulation to a target value of 3,000,000, and a maximum injection time of 50 ms, followed by higher-energy collision dissociation (HCD) fragmentation on the six most intense signals per cycle. HCD spectra were acquired at a resolution of 120,000 using a normalized collision energy of 28 and a maximum injection time of 247 ms. The automatic gain control was set to 100,000 ions. Charge state screening was enabled. Singly, unassigned and charge states higher than six were rejected. Only precursors with intensity above 18,000 were selected for MS/MS. Precursor masses previously selected for MS/MS measurement were excluded from further selection for 30 s, and the exclusion window was set at 10 ppm. The samples were acquired using internal lock mass calibration on $m/z$ 371.1012 and 445.1200.

For samples with laboratory identifiers starting with DA (Supplementary Table 3), mass spectrometry analysis was performed on a Q Exactive mass spectrometer (Thermo Scientific) equipped with a Digital PicoView source (New Objective) and coupled to a nanoAcquity UPLC (Waters). Solvent composition at the two channels was 0.1% formic acid for channel A and 0.1% formic acid, 99.9% ACN for channel B. Column temperature was 50 °C. For each sample, 4 μl of peptides were loaded on a commercial ACQUITY UPLC M-Class Symmetry C18 Trap column (100 Å, 5 μm, 180 μm × 20 mm, Waters) followed by ACQUITY UPLC M-Class HSS T3 column (100 Å, 1.8 μm, 75 μm × 250 mm, Waters). The peptides were eluted at a flow rate of 300 nl min⁻¹ by a gradient from 8 to 22% B in 49 min and to 32% B in additional 11 min. Column was cleaned after the run by increasing to 95% B and holding 95% B for 5 min before re-establishing the loading condition. Samples were acquired in a given order.

The mass spectrometer was operated in data-dependent mode, acquiring a full-scan mass spectra (300–1,700 $m/z$) at a resolution of 70,000 at 200 $m/z$ after accumulation to a target value of 3,000,000, and a maximum injection time of 110 ms followed by HCD fragmentation on the 12 most intense signals per cycle. HCD spectra were acquired at a resolution of 35,000 using a normalized collision energy of 25 and a maximum injection time of 110 ms. The automatic gain control was set to 50,000 ions. Charge state screening was enabled. Singly, unassigned and charge states higher than seven were rejected. Only precursors with intensity above 9,100 were selected for MS/MS. Precursor masses previously selected for MS/MS measurement were excluded from further selection for 30 s, and the exclusion window was set at 10 ppm. The samples were acquired using internal lock mass calibration on $m/z$ 371.1012 and 445.1200.

As all samples in our study were digested with trypsin, peptides had either an arginine or lysine at the C terminus. This resulted in the C-terminal fragments remaining charged, and therefore identified at a higher intensity than b-ions (Extended Data Fig. 1). The mass spectrometry proteomics data were handled using the local laboratory information management system[51] and all relevant data have been deposited to the ProteomeXchange Consortium via the PRIDE (http://www.ebi.ac.uk/pride) partner repository.

**Data analysis.** To account for as much variation of milk-associated proteins as possible during MS/MS ion searches, a supplementary database of milk protein sequences that had not been reviewed was curated from UniProtKB in addition to those from ancient horses, as previously generated[19]. As a previous publication[19], peak lists were generated from raw files by selecting the top 100 peaks using MSConvert from the ProteoWizard software package version 3.0.11781[52]. Sample analysis results were searched using Mascot[53] (version 2.6.0) against the Swiss-Prot database in combination with a curated milk protein database[19]. Results were exported from Mascot as .csv files, and further processed through an internally created tool, MS-MARGE[19,54], to estimate the validity of peptide identifications and summarize the findings. False-discovery rates at both the peptide spectral match and protein

level were calculated using MS-MARGE by counting the number of decoy hits after filtering for $e$-value and minimum peptide support, then dividing this value by the number of target hits minus the number of decoys. The resulting value is multiplied by 100 to provide an estimate of the false-discovery rate. For each individual sample, we aimed for a protein false-discovery rate of under 5% and a peptide false-discovery rate of under 2% (Supplementary Table 4). A minimum of two individual peptide spectral matches were required for each specific protein identification, and only peptide spectral matches with an $e$ value of below 0.01 were accepted. After filtering criteria were applied, we observed a range of variation in the numbers of proteins identified, with samples ranging from 25 to 196 confidently identified protein families.

**Sequence similarities between casein and *Jeotgalicoccus*.** During necessary BLAST searches to authenticate the taxonomic specificity of ruminant α-S1 casein peptides, we found identical sequence matches to theoretical proteins for the numerous bacterial firmicute species from the genus *Jeotgalicoccus* (NCBI reference sequence: WP_188349304.1). Upon further investigation, the full amino acid sequence for these hypothetical bacterial proteins is almost identical to ruminant casein sequences, which is probably due to laboratory contamination during the genomic sequencing. As its listing in the NCBI database is not associated with a publication, we assume this is probably contamination. Supplementary Figure 2 shows the alignment comparing the α-S1 casein sequence for *Bos taurus*, *Bos grunniens*, *Bubalus bubalis* and *Jeotgalicoccus* species.

**Proteome preservation assessment with the Oral Signature Screening Database.** To confirm the preservation of the calculus from individuals included in this study, the metaproteome from each sample was examined for a combination of specific protein types. Following a previous publication[6] we compared the data from each sample against the Oral Signature Screening Database (OSSD) to determine the number of common laboratory contaminants, contaminants introduced during handling and curation, regularly recovered human immune proteins found in the oral cavity, and bacterial proteins common to the human oral microbiome. Supplementary Table 3 contains the overall count of OSSD proteins pulled from our filtered results, as well as the result of the oral microbiome protein identifiers + human immune proteins divided by the total number of OSSD proteins multiplied by 100 to find the 'authenticity' of oral signature proteins in comparison to the total proteins recovered. To determine who among the individuals passed our screening, we applied a different threshold to each time period. For the Early and Middle Bronze Age, we applied a previously published standard[37], and for the Eneolithic period samples we lowered the standard to 40% to take into account increased protein degradation over time. Individuals who had calculus that fell below the authenticity threshold were excluded from the study, but remain listed on the preservation table. Sample authenticity is further supported by an absence of dietary proteins in all positive (archaeological sheep bone with known proteome) and negative controls (extraction blank), as well as the fact that none of the control samples showed any evidence of a typical oral protein signature. Protein preservation varies greatly between different environments and can even differ between individuals at the same site[37,55], and this assessment should be conducted on a project-by-project basis.

**Bayesian estimates of dietary contributions from freshwater protein and radiocarbon calibration adjusted for freshwater dietary radiocarbon reservoir effects for Eneolithic individuals.** Chronologies based on human radiocarbon dates require estimates of individual aquatic dietary intakes, as well as estimates of aquatic radiocarbon reservoir effects of consumed aquatic protein[56,57]. For the latter, we considered a wide potential variability of between 0 and 1,000 years, which covers previously reported archaeological measurements of coeval terrestrial and aquatic samples and the majority of measurements made

on modern freshwater species from our study region[58]. To estimate the dietary contributions from aquatic protein we used the Bayesian mixing model ReSources developed within the Pandora & IsoMemo initiatives (https://isomemoapp.com/). ReSources is a R-based model that follows a similar implementation to the Bayesian mixing model FRUITS[59]. We defined a two-end member model (terrestrial versus freshwater animal protein) with stable nitrogen reference values for these ($\delta^{15}N_{terrestrial} = 7.1 \pm 2‰$, $\delta^{15}N_{freshwater} = 10.6 \pm 1‰$) calculated following a literature review of previously reported values for bone collagen extracted from terrestrial and freshwater animal species within the study region[26,60]. As with previous similar models, protein reference values are corrected for offsets between bone collagen and edible meat, and the implemented model also included a dietary to consumer isotopic offset[56,57]. For each human bone collagen $\delta^{15}N$ value, ReSources provided an estimate (expressed as a mean and s.d.) of the dietary intake of freshwater protein. This estimate was included within the Bayesian chronological model OxCal v.4.4 to express the degree of mixing between the terrestrial radiocarbon calibration curve IntCal20 and a freshwater radiocarbon curve[61,62]. The latter was defined from IntCal20 by adding a uniform prior of between 0 and 1,000 years. Calibrated radiocarbon dates for each individual are expressed as 95% credible intervals. An example of the OxCal code is given below.

```
Plot()
{
Curve("IntCal20","IntCal20.14c");
Curve("FRE","IntCal20.14c");
Delta_R("LocalFRE", U(0,1000));
Mix_Curves("Date1", "IntCal20","LocalFRE", 63,26);
R_Date("OxA-35976", 5965, 20);
Mix_Curves("Date2", "IntCal20","LocalFRE", 36,22);
R_Date("OxA-37350", 4390, 20);
};
```

### Radiocarbon sample preparation methods

Bone sample preparation methods for radiocarbon data followed previously described methods[63]. In brief, the outer bone surfaces were removed manually and all samples were soaked in successive washes of methanol, acetone and dichloromethane for 30 min each at room temperature to remove adhesives and consolidants, and rinsed in >18.2 MΩ cm⁻¹ water. Bones were demineralized in 0.5 N HCl for 24-36 h at 5 °C, and then gelatinized in 0.01 N HCl for 12 h at 60 °C. On the basis of crude gelatin yield and quality, the gelatin was either ultrafiltered (30-kDa MWCO), or hydrolysed for XAD purification. Resulting material was then combusted under vacuum in sealed quartz tubes with CuO and Ag wire, and the resulting $CO_2$ was converted to graphite using $H_2$ reduction over an iron catalyst. Radiocarbon content was measured on a 500-kV NEC 1.5SDH-1 compact accelerator, and conventional ages were calculated by normalizing to OXII oxalic acid standards and correcting for fractionation using the $\delta^{13}C$ ratio measure on the AMS[64].

### Reporting summary

Further information on research design is available in the Nature Research Reporting Summary linked to this paper.

## Data availability

All raw, peak and result protein data have been uploaded to ProteomExchange (http://www.proteomexchange.org). Files are available under the project accession: PXD022300, and the project DOI is https://doi.org/10.6019/PXD022300. S.W. can also be contacted at shevan.wilkin@iem.uzh.ch.

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

**Acknowledgements** We thank the Max Planck Society for providing the funding for this project. D.R. is an Investigator of the Howard Hughes Medical Institute. A.E. was supported by a grant from the Russian Science Federation, grant number 20-18-00402. We thank A. Dittman for insights on protein mass spectra.

**Author contributions** S.W. and N.B. designed the study; S.W., A. Khokhlov, E.K., V.F.Z., A.K.O., D.R.B., P.K., A. Kitova and A.E. participated in sample collection; S.W. conducted the protein extractions and analysed the data; L.K. and C.F. ran the samples on the LC–MS/MS; D.R., R.F., B.J.C., D.K., A. Khokhlov and E.K. provided radiocarbon dates; S.W. and N.B. wrote the draft with the help of A.V.M., W.T.-T.T., R.S., D.A., E.K. and A. Khokhlov, and with input from all other co-authors.

**Funding** Open access funding provided by Max Planck Society.

**Competing interests** The authors declare no competing interests.

**Additional information**
**Correspondence and requests for materials** should be addressed to Shevan Wilkin or Nicole Boivin.

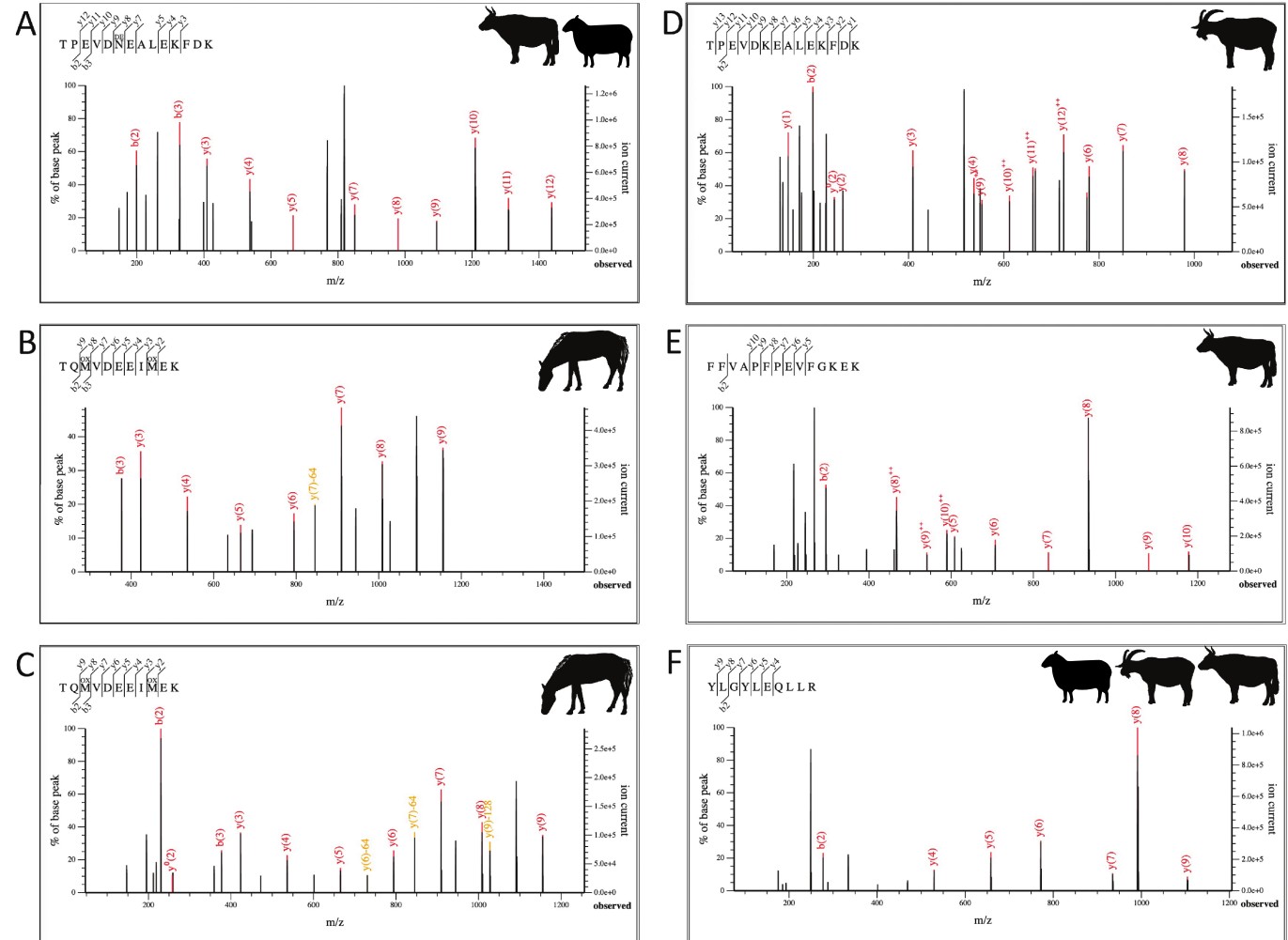

**Extended Data Fig. 1 | MS2 spectra for dairy proteins. a**, BLG peptide specific to *Ovis* or Bovinae for DA420. **b**, BLG I peptide specific to *Equus* for DA420. **c**, *Equus* BLG I peptide for Z438. **d**, MS2 spectra for a *Capra*-specific BLG peptide for Z438. **e**, α-S1 casein from DA430 specific to Bovinae. **f**, Second α-S1 casein peptide specific to Bovidae, also from DA430. Horse, goat and cow images are reproduced from ref. [37]; sheep silhouette, public domain (https://thenounproject.com/icon/12538/).

# nature research

Nicole Boivin

# Reporting Summary

Nature Research wishes to improve the reproducibility of the work that we publish. This form provides structure for consistency and transparency in reporting. For further information on Nature Research policies, see Authors & Referees and the Editorial Policy Checklist.

## Statistics

For all statistical analyses, confirm that the following items are present in the figure legend, table legend, main text, or Methods section.

| n/a | Confirmed | |
|---|---|---|
| ☐ | ☒ | The exact sample size (*n*) for each experimental group/condition, given as a discrete number and unit of measurement |
| ☒ | ☐ | A statement on whether measurements were taken from distinct samples or whether the same sample was measured repeatedly |
| ☒ | ☐ | The statistical test(s) used AND whether they are one- or two-sided *Only common tests should be described solely by name; describe more complex techniques in the Methods section.* |
| ☒ | ☐ | A description of all covariates tested |
| ☒ | ☐ | A description of any assumptions or corrections, such as tests of normality and adjustment for multiple comparisons |
| ☒ | ☐ | A full description of the statistical parameters including central tendency (e.g. means) or other basic estimates (e.g. regression coefficient) AND variation (e.g. standard deviation) or associated estimates of uncertainty (e.g. confidence intervals) |
| ☒ | ☐ | For null hypothesis testing, the test statistic (e.g. *F*, *t*, *r*) with confidence intervals, effect sizes, degrees of freedom and *P* value noted *Give P values as exact values whenever suitable.* |
| ☒ | ☐ | For Bayesian analysis, information on the choice of priors and Markov chain Monte Carlo settings |
| ☒ | ☐ | For hierarchical and complex designs, identification of the appropriate level for tests and full reporting of outcomes |
| ☒ | ☐ | Estimates of effect sizes (e.g. Cohen's *d*, Pearson's *r*), indicating how they were calculated |

*Our web collection on statistics for biologists contains articles on many of the points above.*

## Software and code

Policy information about availability of computer code

| Data collection | Mass spectrometry analysis was performed on a Q Exactive HF mass spectrometer (Thermo Scientific) equipped with a Digital PicoView source (New Objective) and coupled to a M-Class or nanoAcquity UPLC (Waters). |
|---|---|
| Data analysis | To transfer raw MS/MS files to Mascot Generic Files (mgf) we used the program MSConvert from the ProteoWizard software package version 3.0.11781. Then, data were searched using Mascot (version 2.6.0) and filtered using MS-MARGE (https://bitbucket.org/rwhagan/ms-marge/src/master/) |

For manuscripts utilizing custom algorithms or software that are central to the research but not yet described in published literature, software must be made available to editors/reviewers. We strongly encourage code deposition in a community repository (e.g. GitHub). See the Nature Research guidelines for submitting code & software for further information.

## Data

Policy information about availability of data

All manuscripts must include a data availability statement. This statement should provide the following information, where applicable:
- Accession codes, unique identifiers, or web links for publicly available datasets
- A list of figures that have associated raw data
- A description of any restrictions on data availability

All raw, peak, and result protein data has been uploaded to ProteoExchange (http://www.proteomexchange.org). Files are available under the project accession: PXD022300, and the project DOI is 10.6019/PXD022300.

# Field-specific reporting

Please select the one below that is the best fit for your research. If you are not sure, read the appropriate sections before making your selection.

☒ Life sciences ☐ Behavioural & social sciences ☐ Ecological, evolutionary & environmental sciences

For a reference copy of the document with all sections, see nature.com/documents/nr-reporting-summary-flat.pdf

# Life sciences study design

All studies must disclose on these points even when the disclosure is negative.

| | |
|---|---|
| Sample size | Sample size was determined by the number of archaeological individuals that had accumulated dental calculus available for collection. |
| Data exclusions | Dental calculus samples that did not pass the preservation threshold were excluded from further analysis, however, these samples are still listed and their preservation score is provided in Supplementary Table S3. |
| Replication | These findings can be easily replicated by downloading either the 'raw' or 'mgf' files and re-searching them against the same databases. |
| Randomization | Samples were not randomized as it was not necessary for the type of study we conducted. |
| Blinding | As the individuals were not tested for any sort of reaction or non-reaction blinding was not necessary for our study. |

# Reporting for specific materials, systems and methods

We require information from authors about some types of materials, experimental systems and methods used in many studies. Here, indicate whether each material, system or method listed is relevant to your study. If you are not sure if a list item applies to your research, read the appropriate section before selecting a response.

### Materials & experimental systems

| n/a | Involved in the study |
|---|---|
| ☒ | ☐ Antibodies |
| ☒ | ☐ Eukaryotic cell lines |
| ☐ | ☒ Palaeontology |
| ☒ | ☐ Animals and other organisms |
| ☒ | ☐ Human research participants |
| ☒ | ☐ Clinical data |

### Methods

| n/a | Involved in the study |
|---|---|
| ☒ | ☐ ChIP-seq |
| ☒ | ☐ Flow cytometry |
| ☒ | ☐ MRI-based neuroimaging |

## Palaeontology

| | |
|---|---|
| Specimen provenance | Dental calculus samples were collected from Samara State University Department of Archaeology Collection Saplesand the scientific collections of the Museum at the Institute of Plant and Animal Ecology (Ural Branch of the Russian Academy of Sciences). |
| Specimen deposition | ProteomExchagne http://www.proteomexchange.org/ login: reviewer_pxd022300@ebi.ac.uk and password of: 7TwobDNV |
| Dating methods | AMS radiocarbon dating was conducted to dates some individuals. Bone sample preparation methods for radiocarbon data follow those as described in Narasimhan et al. Briefly, the outer bone surfaces were removed manually and all samples were soaked in successive washes of methanol, acetone and dichloromethane for 30 min each at room temperature to remove adhesives and consolidants and rinsed in >18.2 MΩ/cm water. Bones were demineralized in 0.5N HCl for 24-36 hr at 5°C, and then gelatinized in 0.01N HCl for 12hr at 60°C. Based on crude gelatin yield and quality, the gelatin was either ultrafiltered (30k Da MWCO), or hydrolyzed for XAD purification. Resulting material was then combusted under vacuum in sealed quartz tubes with CuO and Ag wire, and the resulting $CO_2$ was converted to graphite using $H_2$ reduction over an iron catalyst. Radiocarbon content was measured on a 500kV NEC 1.5SDH-1 compact accelerator, and conventional ages were calculated by normalizing to OXII oxalic acid standards and correcting for fractionation using the $\delta 13C$ ratio measure on the AMS15. |

☒ Tick this box to confirm that the raw and calibrated dates are available in the paper or in Supplementary Information.

