## [Peer Review File · Nature]

Manuscript Title: Dairying enabled Early Bronze Age Yamnaya steppe expansions

Reviewer Comments & Author Rebuttals

Reviewer Reports on the Initial Version:

Referees' comments:

Referee #1 (Remarks to the Author):

In this manuscript, Wilkin et al. analyze enamel from humans across the Eurasian steppe and find a transition from non-dairy individuals to dairy consumption in the Early Bronze Age. Their results have implications for evaluating the domestication of animals at this time.

Specific comments:

Page 4: " Sample authenticity is further supported by an absence of dietary proteins in all positive and negative controls, as well as the fact that none of the control samples showed any evidence of a typical oral protein signature." What were the positive controls?

Page 4: " Additionally, within the two identified casein peptides, there is only one possible amino acid

deamidation site, rendering any estimation of the antiquity of these peptides exceedingly challenging.

It is thus difficult to confirm the authenticity of this dairy finding." After this paper was submitted, Ramsøe et al. cautioned that deamidation should not be used for antiquity measurements because of high variability of deamidation in milk peptides. Please revise this sentence to reflect this caution.

Ramsøe, A.; Crispin, M.; Mackie, M.; McGrath, K.; Fischer, R.; Demarchi, B.; Collins, M. J.; Hendy, J.; Speller, C., Assessing the degradation of ancient milk proteins through site-specific deamidation patterns. *Scientific Reports* 2021, 11, (1), 7795.

Page 5: "Pecora, or all even-toed ruminants" Why not use Artiodactyla instead of even-toed ruminants, especially after using Pecora? Please revise.

Figure 2B: Pecora is cropped. Please revise.

Fig. 3: All of these annotations are quite difficult to read. Please make this figure larger or split it between the main text and supplement. Also, why does it appear there are no/few b-series ions?

Page 15: "Contamination from modern human collagen" Human collagen (of any type) isn't a contaminant unless you cut yourself or have modern human extracellular matrix in the same lab. Do you mean keratin that can be transferred from the hands?

Page 15 Protein extraction: how were samples decontaminated

Page 15: "5-7 days until completion" What temperature was used? What centrifuge speed was used. Different centrifuges have different top speeds.

Page 16: " To increase denaturation, reduce and alkylate t200 ul of 6 M guanidine hydrochloride and 30 ul of 40 mM CAA, 100 mM TCEP were added to the pellet and remaining supernatant and

mixed through resuspension. Samples were then heated to 99°C for 10 minutes." It looks like this sentence was shortened or partially removed. Please revise for clarity.

Page 17: "The mass spectrometer..." All values above 1000 have apostrophes instead of commas, please revise.

Page 17: What is the ProteomeXchange number?

Page 18 Jeotgalicoccus: This seems like a contaminant in the Jeotgalicoccus genomic sequencing. On NCBI there is no associated paper with the sequence. There would be no reason for a 100% match to a cow milk protein unless it was present in the cell culture used for genomic sequencing. TGY medium contains tryptic casein, so it is possible that this is the origin of the casein sequence. Please consider adding a comment that it is likely contamination in the NCBI database.

Page 18 Figure 4: Is this figure missing?

Supplementary tables/Supplement: Please add which museum these individuals will be housed in. I appreciate the inclusion of the sample numbers, but it is unclear where they are post-excavation.

Referee #2 (Remarks to the Author):

The expansion of Proto-Indo-European (PIE) is an important event in human history during mid-Holocene, which had a great impact on the development of human society. For a long time, there is lack of hard evidence for the expansion mechanism. This study successfully identified the species of milk in human dental calculus from different periods in the Pontic-Caspian steppe by using proteomics. It seems that "Secondary Production Revolution" happened around 3300BC with subsistence shift from early reliance on fishing-hunting to more reliance on dairy animals. More critically, the first finding of horse milk peptides in the samples showed another horse domestication center here, and the horse exploitation would surely support mobility. So this work is a base to understand the expansion of Yamnaya cultural horizon. Thus, the study deserves publication.

Some advice are as follows:

- 1) The authors discussed the subsistence shift in the Pontic steppe around 3300 BC with the bone isotope analysis. Is there more zooarchaeological data reflecting this shift?
- 2) It seems that milking didn't spread with domestic ruminant animals before 3300BC, which is different to the situation in Anatolia, Europe and Africa, where milking and cattle/sheep/goat spread together. Could the authors give some explanation for this?
- 3) After "Secondary Production Revolution" happened around 3300BC, was there some change of social structure or settlement patterns, society complexity?
- 4) The authors compared pottery lipid residue analysis with proteomic analysis during EBA, it's better to list the calibrated calendar age of these pottery residue in the main text.
- 5) In Fig.1, it's better to mark the site without using the names, maybe using the number to indicate and then explain in the caption. Using horse symbols in Fig.1 also bring some confusion, specially no milk peptides were surely identified in the early period.

Author Rebuttals to Initial Comments:

Wilkin et al. – Response to reviewers

We would like to heartily thank both reviewers for their insights and comments, which have helped us to make this a better paper. We outline below how we have adjusted our manuscript to take on board these constructive suggestions. We cite the altered text in this response document, and also track all changes in the manuscript and supplemental text, highlighting in yellow those that are a response to the reviews (minor stylistic changes were made beyond the reviewer comments and these are also shown). Below, we have bolded the reviewer comments for clarity.

Referee #1 (Remarks to the Author):

In this manuscript, Wilkin et al. analyse enamel from humans across the Eurasian steppe and find a transition from non-diary individuals to diary consumption in the Early Bronze Age. Their results have implications for evaluating the domestication of animals at this time.

Specific comments:

Page 4: " Sample authenticity is further supported by an absence of dietary proteins in all positive and negative controls, as well as the fact that none of the control samples showed any evidence of a typical oral protein signature." What were the positive controls?

Authors' response: Thank you for pointing out that we neglected to detail our positive control. The following text has been added to reflect that we employed archaeological sheep bone as a positive control:

"Sample authenticity is further supported by an absence of dietary proteins in all positive (archaeological sheep bone with known proteome) and negative controls (extraction blank), as well as the fact that none of the control samples showed any evidence of a typical oral protein signature."

Page 4: " Additionally, within the two identified casein peptides, there is only one possible amino acid deamidation site, rendering any estimation of the antiquity of these peptides exceedingly challenging. It is thus difficult to confirm the authenticity of this dairy finding." After this paper was submitted, Ramsøe et al. cautioned that deamidation should not be used for antiquity measurements because of high variability of deamidation in milk peptides. Please revise this sentence to reflect this caution.

Ramsøe, A.; Crispin, M.; Mackie, M.; McGrath, K.; Fischer, R.; Demarchi, B.; Collins, M. J.; Hendy, J.; Speller, C., Assessing the degradation of ancient milk proteins through site-specific deamidation patterns. *Scientific Reports* 2021, 11, (1), 7795.

Authors' response: This is an important paper, and we are very happy to be able to reference it in our manuscript. Here is the text addition:

"A recently published paper⁴³ demonstrates the extreme variability in deamidation of amino acids in milk proteins, further limiting our ability to confirm the authenticity of this dairy finding."

Page 5: "Pecora, or all even-toed ruminants" Why not use Artiodactyla instead of even-toed ruminants, especially after using Pecora? Please revise.

Authors' response: We agree and have revised accordingly:

"While many milk peptides were only specific to higher taxonomic levels, such as Pecora, an infraorder within Artiodactyla (cow, sheep, goat, buffalo, yak, reindeer, deer, and antelope), others enabled more specific taxonomic classifications, such as to family, genus, or species."

Figure 2B: Pecora is cropped. Please revise.

Authors' response: Thank you for this observation. We have further rotated the charts 90 degrees to increase readability of the sample names.

Fig. 3: All of these annotations are quite difficult to read. Please make this figure larger or split it between the main text and supplement. Also, why does it appear there are no/few b-series ions?

Authors' response: We agree that this figure is a bit small, and we have rearranged and enlarged the individual spectra figures to make them more readable.

The scarcity of the b-series ions is a valid question, and reflects the type of machine used, as well as the C-terminus ionization of tryptic peptides. For MS/MS analysis, we used an 'orbitrap' machine (Thermo Fisher Q-Exactive), which generally recovers more y-ions than an 'ion trap', which results in more b-ions. Also, as our peptides were digested with trypsin, they end with the basic amino acids arginine (R) or lysine (K), resulting in the charge being retained at the C-terminus. This causes the C-terminus end of the peptide to "fly" better in the machine, and y-ions are measured at a greater intensity than the b-ions.

We have added the following brief text to the MS/MS analysis methods section to explain this:

"As all samples in our study were digested with trypsin, peptides had either an arginine (R) or lysine (K) at the C-terminus. This resulted in the C-terminus fragments remaining charged and these are therefore identified at a higher intensity than b-ions (see Figure 3 A-F)."

Page 15: "Contamination from modern human collagen" Human collagen (of any type) isn't a contaminant unless you cut yourself or have modern human extracellular matrix in the same lab. Do you mean keratin that can be transferred from the hands?

Authors' response: Most of the collagen, we assume, is passed from the skeletal remains themselves during excavation and curation. The calculus we sampled was from individuals excavated over the last 10 years and stored in separate boxes, but in the same room with many other skeletons. Also, many have been sampled for DNA and stable isotope analysis, which often leads to the creation of bone powder in their storage environment. This is

where we assume that the collagen contamination comes from. We have adjusted the text to better clarify this:

“Contamination from modern human keratin and environmental collagen (from the storage environment) was reduced through the use of nitrile gloves during collection.”

Page 15 Protein extraction: how were samples decontaminated

Authors’ response: We have added the following text to explain this in the “Z” sample designation paragraph:

“Z”: “To decrease contamination, one millilitre of 0.5M EDTA was then added to each sample tube. Samples were rotated for 5 minutes followed by centrifugation at 20,000 rcf for 10 minutes to remove the any contamination on the outer layer of calculus, and the supernatant was then removed and retained. Then a second millilitre of 0.5 M EDTA was added to each decontaminated sample, and the sample was allowed to decalcify under rotation for 5-7 days until completion.”

Page 15: "5-7 days until completion" What temperature was used? What centrifuge speed was used. Different centrifuges have different top speeds.

Authors’ response: We have now provided these details in the text:

“Then a second millilitre of 0.5 M EDTA was added to each decontaminated sample, and the sample was allowed to decalcify under rotation for 5-7 days at room temperature until completion. After demineralization, samples were centrifuged at top speed (20,000 rcf) for 5 minutes.”

Page 16: " To increase denaturation, reduce and alkylate t200 ul of 6 M guanidine hydrochloride and 30 ul of 40 mM CAA, 100 mM TCEP were added to the pellet and remaining supernatant and mixed through resuspension. Samples were then heated to 99°C for 10 minutes." It looks like this sentence was shortened or partially removed. Please revise for clarity.

Authors’ response: We have altered the text to be clearer:

“Samples were then placed on a heating block (Cell Media, Thermoshaker Pro) and heated to 99°C for 10 minutes.”

Page 17: "The mass spectrometer..." All values above 1000 have apostrophes instead of commas, please revise.

Authors’ response: This has been adjusted to properly include commas rather than apostrophes.

Page 17: What is the ProteomeXchange number?

Authors' response: We have added the information, as well as removed the reviewer login information from the main text and replaced it with the final accession and DOI as to be accessible by the public.

"Files are available under the project accession: PXD022300, and the project DOI is 10.6019/PXD022300."

Page 18 Jeotgalicoccus: This seems like a contaminant in the Jeotgalicoccus genomic sequencing. On NCBI there is no associated paper with the sequence. There would be no reason for a 100% match to a cow milk protein unless it was present in the cell culture used for genomic sequencing. TGY medium contains tryptic casein, so it is possible that this is the origin of the casein sequence. Please consider adding a comment that it is likely contamination in the NCBI database.

Authors' response: We were thinking along those lines, and appreciate the clarification on this point. We have adjusted the text to better explain the likely contamination in the NCBI database:

"Upon further investigation, the full amino acid sequence for these hypothetical bacterial proteins is almost identical to ruminant casein sequences, which is likely due to laboratory contamination during the genomic sequencing. As its listing in the NCBI database is not associated with a publication, we assume this is likely contamination."

Page 18 Figure 4: Is this figure missing?

Authors' response: Yes, this figure is missing, and has now been added.

Supplementary tables/Supplement: Please add which museum these individuals will be housed in. I appreciate the inclusion of the sample numbers, but it is unclear where they are post-excavation.

Authors' response: We thank the reviewer for pointing out this important omission. We have added the following text to the supplement:

"Individuals were sampled from three different institutions: All individuals except those from Kammenyi Ambar 5 and Botai are curated at Samara State University's Department of Archaeology. The three individuals from the site of Kammenyi Ambar 5 belong to the scientific collections of the Museum at the Institute of Plant and Animal Ecology (Ural Branch of the Russian Academy of Sciences). The two samples collected from individuals from the site of Botai were sampled by Alan Outram and delivered to MPI-SHH, the individuals are stored by Victor F. Zaubert at the Botai Research Station."

Referee #2 (Remarks to the Author):

The expansion of Proto-Indo-European (PIE) is an important event in human history during mid-Holocene, which had a great impact on the development of human society. For a long

time there is lack of hard evidence for the expansion mechanism. This study successfully identified the species of milk in human dental calculus from different periods in the Pontic-Caspian steppe by using proteomics. It seems that “Secondary Production Revolution” happened around 3300BC with subsistence shift from early reliance on fishing-hunting to more reliance on dairy animals. More critically, the first finding of horse milk peptides in the samples showed another horse domestication center here, and the horse exploitation would surely support mobility. So this work is a base to understand the expansion of Yamnaya cultural horizon. Thus, the study deserves publication.

Some advice are as follows:

1) The authors discussed the subsistence shift in the Pontic steppe around 3300 BC with the bone isotope analysis. Is there more zooarchaeological data reflecting this shift?

Authors’ response: This is an excellent question. Zooarchaeological remains are difficult to compare between the Eneolithic and Yamnaya periods because most Eneolithic settlements (with fauna) were abandoned during the Yamnaya period. Furthermore, in the study area, no Eneolithic settlements show occupation into the Yamnaya era, a pattern that is ascribed to the extreme mobility of the Yamnaya population. However, there are a couple of Yamnaya settlements in the Dnieper Valley with fauna (Mikhailovka II-III & Generalka), both contain 85-90% domesticated species, specifically cattle, sheep-goat, and horse, with cattle dominating (66% - 78%). This represents a shift from quite variable Eneolithic riverine settlement faunal assemblages that generally had more horses and deer than cattle or sheep-goat, and that also contained abundant fish remains. Stable isotopes in human bone are a more widely available source of information on dietary shift since they can be studied in regions without Yamnaya settlements, and they are referenced in the new text:

“The proteomic data agree with stable isotope analysis of individuals from Eneolithic to Bronze Age Samara showing a correspondent shift from a heavy reliance on fish, deer, and other riverine forest (C3) resources to a greater reliance on terrestrial and grassland (C3 & C4) animal products^{21,45}. Eneolithic riverine settlements, where subsistence focused on a mixture of domesticated and wild animals and fish, were abandoned during the Early Bronze Age and Yamnaya zooarchaeological remains in the Volga-Ural study area are limited to occasional sacrifices in graves, 70-90% of which were sheep-goat⁵². In the Dnieper Valley, two Yamnaya settlements had primarily cattle, followed by sheep-goat and horse^{45, 48, 49}.

2) It seems that milking didn’t spread with domestic ruminant animals before 3300BC, which is different to the situation in Anatolia, Europe and Africa, where milking and cattle/sheep/goat spread together. Could the authors give some explanation for this?

Authors’ response: We agree that this is an interesting difference, and we suggest that a primary factor is that the movement of domestic animals in Anatolia, Africa, and Europe related to the spread of farming populations, who were also dairying. In contrast, domesticated animals in the Pontic Caspian region were adopted by hunter-gatherers. This likely created a cultural frontier between the milking populations of Eastern Europe and the fishers/hunter/gatherers living at the western steppe horizon. We have added the following text to clarify this:

“Interestingly, while neighboring Eneolithic farming populations in Europe appear to have been dairying⁴⁴, those living across the steppe frontier did not adopt milking practices, suggesting the presence of a cultural frontier.”

3) After “Secondary Production Revolution” happened around 3300BC, was there some change of social structure or settlement patterns, society complexity?

Authors’ response: There were dramatic changes in the structural and settlement changes after the Secondary Products Revolution that occurred with the Yamnaya cultural horizon. We have added the following text to better detail these changes:

“This change in subsistence economy, indicated by dietary stable isotopes in human bones as well as by proteomics, was accompanied by the widespread abandonment of Eneolithic riverine settlement sites, the appearance of kurgan cemeteries in the previously unexploited arid plateaus between the river valleys, and the inclusion of wheeled vehicles and occasional horse bones in Yamnaya graves. At the same time, the steppe Yamnaya population expanded dramatically westward into Europe and eastward to the Altai Mountains, a range of 6000 km^{1,3,65}”

4) The authors compared pottery lipid residue analysis with proteomic analysis during EBA, it’s better to list the calibrated calendar age of these pottery residue in the main text.

Authors’ response: Thank you for this recommendation, we agree and have added the date range (of ca. 3800-3000 BCE) to the main text.

5) In Fig.1, it’s better to mark the site without using the names, maybe using the number to indicate and then explain in the caption. Using horse symbols in Fig.1 also bring some confusion, specially no milk peptides were surely identified in the early period.

Authors’ response: The symbol in the figures is a cow, but we can see how that could be unclear, as it does closely resemble the horse symbol. We have accordingly replaced it with a sheep, which should be more clearly discernible from the horse. The horse should only be at one of the EBA sites. Also, we have made the sheep in the early Eneolithic period grey to indicate that the milk identification is rather equivocal.

Reviewer Reports on the First Revision:

Referee #1 (Remarks to the Author):

I thank the authors for their thoughtful responses to the comments.

Two additional points:

While b-series ions are less common than y-series, there are typically more than 1 or 2 b-series ions after HCD. Are the limited b-series possible an artifact of choosing the top 100 peaks in msconvert? The spectra interpretation looks correct, it seems strange to me that there are not more b-ions.

"Contamination from modern human keratin and environmental collagen" I think the explanation that the contamination might come from aDNA and isotope analysis is important to mention in the text. Because that is where the contamination might come from not just bones being in the environment.

Referee #2 (Remarks to the Author):

The expansion of PIE population is an important event in human history. This study offers some evidence for the driving mechanism. The authors answered most concerns in the previous review. The paper looks more suitable for publication. However, a small question should be solved. Line 113, in total 56 samples, 55 samples were successfully extracted. but in Line 123, 19 individuals, just 11 were successfully extracted. Both sentences seem contradictory. The authors also use some ratios, such as 86%, 92%, 94%. It's better to add the original numbers, such as (**/**, 86%).

Author Rebuttals to First Revision:

We again thank both reviewers for their helpful comments. Reviewer comments have been bolded, and our responses are below each.

Referee #1:

Remarks to the Author:

I thank the authors for their thoughtful responses to the comments.

Two additional points:

While b-series ions are less common than y-series, there are typically more than 1 or 2 b-series ions after HCD. Are the limited b-series possible an artifact of choosing the top 100 peaks in msconvert? The spectra interpretation looks correct, it seems strange to me that there are not more b-ions.

We think this may be the case, but it may also be the instrument software on the Orbitrap doing some background filtering and may not appear in the spectrum. We have noticed a similar pattern in other ancient protein papers that have included spectra in their figures, namely Jeong et al 2018, and Scott et al., 2020. As the spectra interpretation does look correct, and can only conjecture on reasons for the less frequent b-ions, we would prefer to keep the text as it is.

"Contamination from modern human keratin and environmental collagen" I think the explanation that the contamination might come from aDNA and isotope analysis is important to mention in the text. Because that is where the contamination might come from not just bones being in the environment.

Yes, we agree that this is important to add, and have included the following text "that may have occurred due to previous sampling for aDNA or stable isotope analysis" to the sentence referred to above.

Referee #2:

Remarks to the Author:

The expansion of PIE population is an important event in human history. This study offers some evidence for the driving mechanism. The authors answered most concerns in the previous review. The paper looks more suitable for publication. However, a small question should be solved.

Line 113, in total 56 samples, 55 samples were successfully extracted. but in Line 123, 19 individuals, just 11 were successfully extracted. Both sentences seem contradictory.

The 56 and 55 are referring to the entirety of the samples from which proteins were successfully extracted. In the next line we mention that 11 of the 19 samples were successful extractions **and** had good preservation. In the next line we mention that of the 55, 48 had acceptable preservation. From the Eneolithic, 18 were successfully extracted, and 7 of those did not have the level of preservation we expect from an authentic oral signature. So while 55/56 were successfully extracted, 7 from the additional samples (all from the Eneolithic) did not have adequate preservation.

The authors also use some ratios, such as 86%, 92%, 94%. It's better to add the original numbers, such as (/**, 86%).**

We agree that this would be clearer and have added the number of individuals as well as the percentage.

nature portfolio